# WAVE: Learning Unified & Versatile Audio-Visual Embeddings with Multimodal LLM

**Changli Tang[1], Qinfan Xiao[1], Ke Mei[2], Tianyi Wang[2], Fengyun Rao[2], Chao Zhang[1†]**
Tsinghua University[1], WeChat Vision, Tencent Inc.[2]
`tcl24@mails.tsinghua.edu.cn, cz277@tsinghua.edu.cn`

## Abstract

While embeddings from multimodal large language models (LLMs) excel as general-purpose representations, their application to dynamic modalities like audio and video remains underexplored. We introduce WAVE (**u**nified & **v**ersatile **a**udio-**v**isual **e**mbeddings), the first LLM-based embedding that creates a unified representation space for text, audio, and video modalities. WAVE employs a novel hierarchical feature fusion strategy and a joint multi-modal, multi-task training approach to enable two key capabilities: any-to-any cross-modal retrieval and the generation of prompt-aware embeddings tailored to user instructions. Experimentally, WAVE sets a new state-of-the-art on the MMEB-v2 video benchmark and achieves superior results in audio and video-to-audio retrieval. Its prompt-aware nature also yields remarkable performance in multimodal question answering, significantly outperforming existing embedding models. Ablation studies validate our joint training strategy, demonstrating improved performance across all modalities. With a newly introduced benchmark for versatile audio-visual learning, WAVE opens up broad possibilities for cross-modal, any-to-any applications. Our code and checkpoints are released at https://github.com/TCL606/WAVE.

## 1 Introduction

Multimodal embeddings, which transform diverse data types such as text, images, video, and audio into a shared representation space, are central to cross-modal search, classification, and recommendation. The prevailing approach employs separate encoders per modality that are aligned in a common space (Radford et al., 2021; Jia et al., 2021; Zhai et al., 2023; Ma et al., 2022; Miech et al., 2020; Xu et al., 2021; Elizalde et al., 2023; Mei et al., 2024; Guzhov et al., 2022; Su et al., 2024; McKee et al., 2023; Chen et al., 2024b). Recently, the success of large language models (LLMs) has catalysed a more integrated paradigm: using a single multimodal LLM (MLLM) to produce embeddings for all modalities jointly. This shift is enabled by increasingly capable MLLMs that can process and reason over images (Liu et al., 2024b;a; Li et al., 2023; Chen et al., 2023), audio (Tang et al., 2024; Gong et al., 2024; 2023; Chu et al., 2024), and video (Li et al., 2025a; Zhang et al., 2025c; Liu et al., 2025b; Zhu et al., 2025; Bai et al., 2025; Li et al., 2025b; Zhang et al., 2025a; Xu et al., 2025; Tang et al., 2025). Consequently, the field is rapidly moving toward using these models to produce potent, versatile multimodal embeddings (Jiang et al., 2025; Meng et al., 2025; Yu et al., 2025a; Zhang et al., 2024; Jiang et al., 2024; Gu et al., 2025; Lin et al., 2025; Liu et al., 2025a).

A unified embedding paradigm built upon MLLMs fully leverages their strengths in semantic understanding and representation. By processing all modalities within a single model, this approach naturally improves cross-modal interoperability and semantic alignment, which benefits downstream tasks such as retrieval. Such a model can also ingest multiple modalities concurrently to form holistic representations—for example, a coherent embedding from paired audio and video streams. Furthermore, by inheriting the instruction-following capabilities of MLLMs, the resulting embeddings can be prompt-aware, conditioning on user instructions to encode task-relevant semantics. Despite these advantages, most MLLM-based embedding efforts have concentrated on vision, particularly static images, while underexploring audio and synchronised audio-visual streams. Consequently, the promise of a truly universal audio-visual embedding space remains largely unrealised.

---

[†]Corresponding author

To address these limitations, we introduce WAVE, a **u**nified & **v**ersatile **a**udio–**v**isual **e**mbedding MLLM. To the best of our knowledge, WAVE is the first model to produce unified embeddings for text, audio, silent video, and synchronised audio–visual inputs. Built on Qwen2.5-Omni (Xu et al., 2025), WAVE projects heterogeneous inputs into a shared semantic space, enabling seamless cross-modal interaction. Experiments confirm that WAVE produces powerful embeddings, achieving state-of-the-art (SOTA) performance on the MMEB-v2 video track (Meng et al., 2025) and excelling at tasks like any-to-any retrieval (e.g., text-to-video, video-to-audio). Moreover, it can generate prompt-aware embeddings for downstream applications like multimodal question answering (QA). Crucially, WAVE maintains or even surpasses the performance of the base Qwen2.5-Omni on multimodal understanding benchmarks, which is notable since most embedding models show a significant decline in these capabilities compared to their foundational MLLMs.

Our main contributions can be summarised as follows:

- **Versatile audio–visual embedding MLLM**: We introduce WAVE, the first audio–visual embedding MLLM capable of producing unified, general-purpose representations for text, audio, silent video, and synchronised audio–visual inputs. By projecting heterogeneous modalities into a single semantic space, WAVE excels at challenging any-to-any retrieval and achieves SOTA performance on the MMEB-v2 video track.

- **Instruction-following for prompt-aware embeddings**: Leveraging the instruction-following ability of its MLLM backbone, WAVE generates prompt-aware multimodal embeddings. Unlike conventional models that yield task-agnostic representations, WAVE can condition embeddings on a user's task-specific prompt, which is reflected in its strong results on embedding-based multimodal QA.

- **Effective architecture**: We propose a hierarchical feature-fusion strategy that aggregates representations from multiple MLLM layers, yielding stable gains on tasks such as multimodal retrieval. In addition, a dual-encoder design for audio captures complementary cues (*e.g.*, speech and environmental sounds), further enhancing the expressiveness of the learned embeddings.

## 2 BACKGROUND

### 2.1 MULTIMODAL REPRESENTATION LEARNING

Multimodal representation learning seeks to construct a shared embedding space in which text, image, audio, and video can be compared and composed. A major milestone is CLIP (Radford et al., 2021), which uses contrastive learning with dual encoders to align images and text at scale. Building on this paradigm, ALIGN (Jia et al., 2021) shows that training on even larger, noisier corpora, exceeding a billion image–text pairs, yields strong gains on retrieval and classification. SigLIP (Zhai et al., 2023) further simplifies and scales training by replacing the standard InfoNCE objective with a sigmoid loss, removing the need for in-batch negatives and improving efficiency.

This contrastive recipe naturally extends to video. X-CLIP (Ma et al., 2022) adapts the dual-encoder design to video–text retrieval with explicit temporal modelling. To better exploit large but noisy web videos, Miech et al. (2020) enhance contrastive learning with noise-contrastive estimation, enabling learning from loosely aligned narration. VideoCLIP (Xu et al., 2021) strengthens discrimination by mining hard negatives via nearest-neighbour retrieval during training.

In audio–language learning, CLAP (Elizalde et al., 2023) aligns audio and text in a joint space, enabling zero-shot audio classification and cross-modal retrieval. To address the scarcity of high-quality paired data, Mei et al. (2024) introduce the WavCaps corpus and build state-of-the-art audio–language retrieval models with HTSAT (Chen et al., 2022a) and BERT (Devlin et al., 2019).

As audio and vision are naturally synchronised and complementary, learning unified audio–visual representations is an important next step. AudioCLIP (Guzhov et al., 2022) generalises CLIP to a trimodal setting (audio, image, text), enabling richer cross-modal transfer. Su et al. (2024) propose a unified framework for audio–visual representation and generation, and subsequent work explores emerging applications such as video-to-music retrieval (McKee et al., 2023; Chen et al., 2024b).

## 2.2 LLM-Based Embedding Models

Pretrained on vast corpora, LLMs exhibit strong semantic understanding and broad world knowledge, motivating their use as text-embedding generators. Two common strategies adapt decoder-only LLMs into embedding models: *last-token pooling*, which takes the hidden state of the end-of-sentence (EOS) token as the sentence embedding, and *mean pooling*, which averages token-level hidden states. Using last-token pooling, Wang et al. (2024) first synthesize training data with proprietary LLMs and then fine-tune target models with a standard contrastive objective, yielding competitive text embeddings without complex pipelines. NV-Embed (Lee et al., 2025a) removes the causal attention mask and introduces a latent attention layer to improve mean pooling. More recently, Gemini Embedding (Lee et al., 2025b), Qwen3 Embedding (Zhang et al., 2025b), and QZhou-Embedding (Yu et al., 2025b) have set new marks on comprehensive benchmarks such as MTEB (Muennighoff et al., 2022) through large-scale training.

In the multimodal setting, researchers extend MLLMs to carve out a unified semantic space across modalities, aiming to produce robust, general-purpose embeddings with a single model. VLM2Vec (Jiang et al., 2025) is an early effort that trains a visual LLM across diverse multimodal embedding tasks; VLM2Vec-V2 (Meng et al., 2025) broadens coverage to video and documents. Zhang et al. (2024) focus on multimodal retrieval, building an MLLM-based universal retriever for images and text, while E5-V (Jiang et al., 2024) shows that training on text pairs alone can still improve image–text retrieval. MM-Embed (Lin et al., 2025) adopts an image-LLM bi-encoder with modality-aware hard-negative mining to mitigate modality bias. Gu et al. (2025) combine textual discriminative distillation with multimodal contrastive learning to construct an image embedding LLM. LamRA (Liu et al., 2025a) offers a general framework that equips visual LLMs with strong retrieval and re-ranking, and CAFe (Yu et al., 2025a) unifies visual representation learning and generation via a contrastive-autoregressive fine-tuning scheme, enabling a single model to excel at both retrieval and image generation.

# 3 Methods

## 3.1 Model Architecture

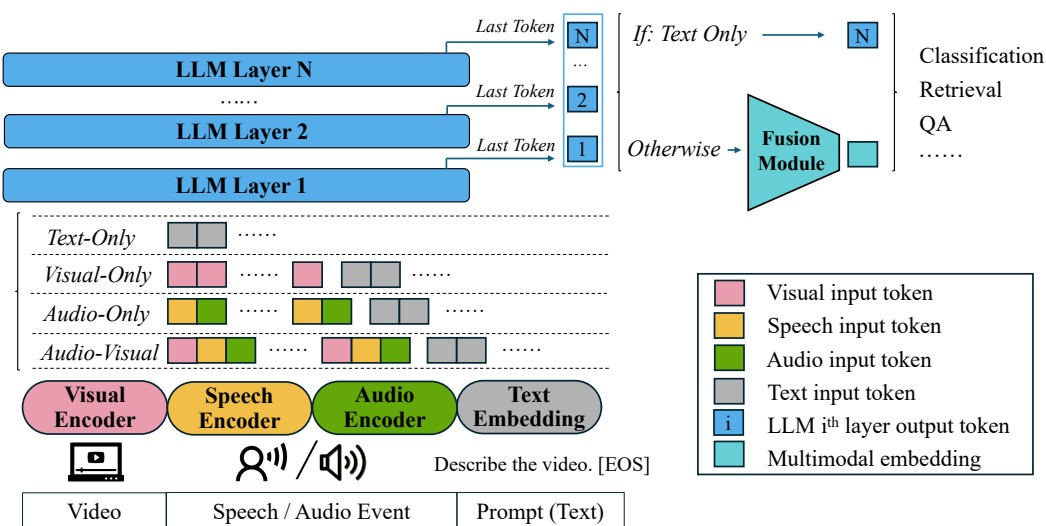

Figure 1: Inputs can be text-only, vision-only, audio-only, or audio–visual. For text-only cases, the final embeddings are obtained via last-token pooling over the LLM's last hidden states. For multimodal inputs, the last output tokens from all LLM layers are concatenated and passed to a feature-fusion module to produce a unified multimodal embedding. Note that text prompts are always provided to instruct the LLM for multimodal inputs.

The overall architecture of WAVE is shown in Fig. 1. The model can accept text, video frames, audio signals, or synchronised audio-visual data as input and generate multimodal embeddings for downstream tasks, such as classification, retrieval, and QA.

To handle this modality diversity, WAVE employs distinct encoders for non-text inputs. A pre-trained visual encoder extracts features from video frames, converting them into visual tokens for the LLM. For audio, we utilise a dual-encoder architecture to comprehensively capture the input signal. A speech encoder and a separate audio encoder generate speech-related and audio event-related tokens, respectively. Text inputs are tokenised using the LLM's original embedding layer. Crucially, all non-text inputs are accompanied by a text prompt, which serves as an instruction to the LLM.

To structure the multimodal input tokens for the LLM, we employ specific interleaving strategies. For audio-only input, the speech-related and audio event-related tokens, which are equal in number due to identical encoder frequencies, are interleaved on a one-to-one basis to form a unified auditory token sequence. For synchronised audio-visual input, both the visual and auditory token sequences are partitioned into several segments corresponding to the number of sampled frames. These segments are then interleaved to create the audio-visual token sequence. Finally, the text tokens of the prompt are appended to the end to form the input token sequence for the LLM.

There are four possible input configurations: text-only, visual-only, audio-only, and audio–visual. Among them, audio and video are both multimodal temporal signals. To enhance the LLM's ability to capture spatiotemporal structure, we adopt the time-aligned multimodal rotary position Embedding (TMRoPE) introduced in Qwen2.5-Omni (Xu et al., 2025). Because the speech and audio encoders are synchronised to produce outputs at the same frequency, their tokens are naturally aligned in time. Tokens corresponding to the same frame, therefore, share the same TMRoPE, ensuring precise temporal alignment.

After TMRoPE is applied, the multimodal token sequence is fed into the LLM. Inspired by the use of the hidden state of the final "EOS" token by last-token pooling, we aggregate the last output tokens from multiple LLM layers to construct embeddings for non-text modalities. This design captures both low-level perceptual cues and high-level semantic abstractions. A lightweight fusion module—implemented as a two-layer multi-layer perceptron (MLP) with GELU activation (Hendrycks & Gimpel, 2016) is then used to refine and compress the embeddings. For the text-only scenario, we retain the standard last-token pooling approach, which prior studies have shown to be highly effective.

## 3.2 TRAINING STRATEGY

Similar to previous work, we adopt contrastive learning as the primary training paradigm to align representations from different modalities into a unified embedding space. The semantic similarity between any two embeddings is quantified using the cosine similarity metric.

Our training regimen is composed of two distinct but complementary tasks: multimodal retrieval and QA. The retrieval task requires the model to extract general multimodal embeddings with a general prompt like "Describe the video", while the QA task requires the model to extract prompt-aware embeddings that well interpret the given question. During training, each sample provides a source-target pair of inputs, denoted as $(s, t)$, which are processed by the model to produce their respective embeddings, $e_s$ and $e_t$. The training is performed on mini-batches of size $N$, and we build a task-aware data sampler that ensures samples in the mini-batch belong to the same task.

**Retrieval Task:** For the retrieval task, $s$ and $t$ belong to different modality types, which can be any of the supported formats: text-only, audio-only, visual-only, or audio-visual. This allows for arbitrary any-to-any cross-modal training. We employ an in-batch negative sampling strategy, where for a given positive pair, all other non-corresponding pairs within the mini-batch are treated as negative samples. To ensure a robust alignment, we compute a symmetric InfoNCE loss.

Specifically, for the $i$ th sample in the mini-batch, when its source embedding $e_{s_i}$ serves as the query and its target embedding $e_{t_i}$ is the positive key, the loss $L_{s_i}$ is formulated as a cross-entropy loss

over the batch:

$$\mathcal{L}_{s_i} = -\log \frac{\exp(\mathrm{sim}(e_{s_i}, e_{t_i})/\tau)}{\sum_{j=1}^{N} \exp(\mathrm{sim}(e_{s_i}, e_{t_j})/\tau)}, \tag{1}$$

where $\mathrm{sim}(\cdot, \cdot)$ denotes the cosine similarity, $\tau$ is the temperature parameter, and the summation in the denominator is over all target embeddings $e_{t_j}$ in the mini-batch. For the retrieval task, the source $s$ and target $t$ are interchangeable. Therefore, the symmetrical scenario is also considered, where $e_{t_i}$ serves as the query and $e_{s_i}$ is the positive key. The symmetrical loss $\mathcal{L}_{t_i}$ is:

$$\mathcal{L}_{t_i} = -\log \frac{\exp(\mathrm{sim}(e_{t_i}, e_{s_i})/\tau)}{\sum_{j=1}^{N} \exp(\mathrm{sim}(e_{t_i}, e_{s_j})/\tau)}. \tag{2}$$

The final retrieval loss for the entire mini-batch is the average of these individual losses over all samples and both directions, ensuring a bidirectional alignment:

$$\mathcal{L}_{\mathrm{Retrieval}} = \frac{1}{2N} \sum_{i=1}^{N} (\mathcal{L}_{s_i} + \mathcal{L}_{t_i}). \tag{3}$$

**Question Answering Task** : For the QA task, the source $s_i$ for the $i$th sample in the mini-batch is a multimodal signal accompanied by a textual prompt that poses a question. The corresponding $t_i$ is a text-only input representing the correct answer. To train the model for this discriminative task, we augment each sample with a set of $n$ incorrect or "distractor" answers, denoted as $\{t'_{i,k}\}_{k=1}^{n}$. The model then extracts embeddings for the correct answer, $e_{t_i}$, and for each incorrect answer, $\{e'_{t_{i,k}}\}_{k=1}^{n}$. The objective is to maximise the probability of selecting the correct answer from the candidate pool. The QA loss for the $i$th sample, $\mathcal{L}_{\mathrm{QA}_i}$, is thus formulated as a cross-entropy loss:

$$\mathcal{L}_{\mathrm{QA}_i} = -\log \frac{\exp(\mathrm{sim}(e_{s_i}, e_{t_i})/\tau)}{\exp(\mathrm{sim}(e_{s_i}, e_{t_i})/\tau) + \sum_{k=1}^{n} \exp(\mathrm{sim}(e_{s_i}, e'_{t_{i,k}})/\tau)}. \tag{4}$$

This objective function effectively trains the model to produce a multimodal query embedding $e_{s_i}$ that is most similar to the embedding of the correct textual answer $e_{t_i}$, while being distant from the embeddings of incorrect answers. The total QA loss for the batch is the average of individual losses:

$$\mathcal{L}_{\mathrm{QA}} = \frac{1}{N} \sum_{i=1}^{N} \mathcal{L}_{\mathrm{QA}_i}. \tag{5}$$

## 4 EXPERIMENTAL SETTINGS

### 4.1 MODEL SPECIFICATIONS

WAVE is built on the 7 billion (B) parameter version of Qwen2.5-Omni (Xu et al., 2025). Specifically, the LLM backbone, the visual encoder, and the speech encoder are all initialised from the pre-trained weights of Qwen2.5-Omni, which allows WAVE to inherit the powerful multimodal perception and reasoning capabilities of the foundation model. For the dedicated audio encoder, we adopt BEATs encoder (Chen et al., 2022b) and further append a trainable aligner to align its output with the LLM's input space. The aligner consists of a two-layer MLP projector with a GELU activation function. To ensure efficient fine-tuning, we employ the low-rank adaptation (LoRA) (Hu et al., 2022) technique on the LLM backbone. The LoRA modules are configured with a rank of 128 and a scaling factor of 2.0. A dropout rate of 0.05 is applied to the LoRA modules during training to mitigate overfitting. To generate the ultimate multimodal embeddings, a two-layer MLP with a GELU activation function serves as the fusion module to fuse features from different layers of the LLM. The temperature $\tau$ of the model is set to 0.01.

As for the model input, videos, in general, are sampled at 2 frames per second, with a maximum of 128 frames sampled. For videos longer than 64 seconds, 128 frames are uniformly sampled to conserve memory resources. The maximum resolution for each frame is 176400 pixels. For audio input, the waveform signals are resampled to 16,000 Hz. Other preprocessing settings are identical to those of Qwen2.5-Omni.

## 4.2 TRAINING SPECIFICATIONS

Before the large-scale contrastive learning, we perform a dedicated pre-training phase for the BEATs aligner. This stage aligns the BEATs encoder with the backbone LLM so that the LLM can interpret BEATs features. Only the aligner's parameters are updated while all other components remain frozen. Given an audio clip and a simple text prompt (for example, 'Please describe the audio'), the model is trained to generate a descriptive audio caption. Training uses audio from WavCaps (Mei et al., 2024), AudioCaps (Kim et al., 2019), and Clotho (Drossos et al., 2020); clips longer than 180 seconds are discarded to avoid memory issues. We train for three epochs on 128 H20 GPUs.

Next, we proceed with the primary training stage as detailed in Section 3.2. Table 2 provides a comprehensive overview of the multimodal tasks, data sources, the modalities of each sample pair $(s, t)$, and the number of data samples used in our training. Notably, we re-annotate the 1 million (M) videos of the Panda-70M dataset (Chen et al., 2024a) using InternVL-2.5-8B (Chen et al., 2024c). Besides, in some datasets, a video may correspond to multiple text captions. To enhance the diversity of text captions, we construct samples that share the same video but differ in text captions for these datasets. The final WAVE model is trained on 192 H20 GPUs for one epoch, and the total training time is approximately 36 hours. We set the learning rate to $2 \times 10^{-5}$, and configure a per-device batch size of 1, resulting in a total batch size of 192. The data sampler is designed to ensure that samples in each training mini-batch are consistent in task types and data sources. The visual aligner and the LoRA module are trainable in this stage, while other modules will stay frozen. For ablation experiments, the training settings are similar except that we only use 128 H20 GPUs for training.

Table 1: An overview of training tasks and data. Four tasks are trained for our models: video-text retrieval, video-QA, video-autio retrieval and audio-text retrieval.

| Task | Data Source | Modalities of $(s, t)$ | # Samples |
|---|---|---|---|
| Video-Text Retrieval | Panda-70M (Chen et al., 2024a) | (visual, text) | 1.0 M |
| | MSVD (Chen & Dolan, 2011) | (visual, text) | 24 K |
| | DiDeMo (Anne Hendricks et al., 2017) | (visual, text) | 8 K |
| | ActivityNet Captions (Krishna et al., 2017) | (visual, text) | 10 K |
| | MSR-VTT (Xu et al., 2016) | (audio-visual, text) | 180 K |
| | VATEX (Wang et al., 2019) | (audio-visual, text) | 260 K |
| | YouCook2 (Zhou et al., 2018) | (audio-visual, text) | 10 K |
| | Shot2Story (Han et al., 2023) | (audio-visual, text) | 530 K |
| Video-QA | LLaVA-Video-178k (Zhang et al., 2025c) | (visual, text) | 100 K |
| Video-Audio Retrieval | AudioSet (Gemmeke et al., 2017) | (audio, visual) | 1.7 M |
| | VGGSound (Chen et al., 2020) | (audio, visual) | 182 K |
| Audio-Text Retrieval | AudioCaps (Kim et al., 2019) | (audio, text) | 49 K |
| | AudioSet-SL (Hershey et al., 2021) | (audio, text) | 108 K |
| | Clotho (Drossos et al., 2020) | (audio, text) | 19 K |
| **Total** | - | - | 4.9 M |

## 4.3 EVALUATION SPECIFICATIONS

To thoroughly assess the performance of WAVE, we have collected and organised a comprehensive suite of evaluation tasks and benchmarks. This collection is designed to systematically measure the quality of the embeddings for each modality and to validate the model's cross-modal alignment capabilities.

Details of the evaluation data and metrics are shown in Table 2. All evaluation tasks are formulated as "query-to-target" retrieval. For video-centric tasks, we adopt the video subset from MMEB-v2 (Meng et al., 2025) as our foundation, and we also augment the evaluations on the recent benchmark LoVR (Cai et al., 2025). In the audio domain, our evaluation encompasses both retrieval and QA tasks as well. Beyond these text-centric scenarios, more challenging tasks such as video-to-audio retrieval and video-to-music retrieval are evaluated, which test the model's ability to map visual semantics to auditory concepts directly within its unified embedding space. More details of the inference procedure for each evaluation task are shown in Appendix B.

Table 2: Details of the evaluation benchmarks. We formulate all tasks as "query-to-target" retrieval,

| Data Source | Task | Subset | (Query, Target) Modalities | Metrics |
|---|---|---|---|---|
| MMEB-v2-Video (Meng et al., 2025) | Classfication | CLS | (visual, text) | Acc% |
| | Video QA | QA | (visual, text) | Acc% |
| | Retrieval | RET | (text, visual/audio-visual) | R@1 |
| | Moment retrieval | MRET | (text, visual) | R@1 |
| LoVR (Cai et al., 2025) | Retrieval | text-to-clip | (text, visual) | R@1 |
| | | theme-to-clip | | R@25 |
| AudioCaps (Kim et al., 2019) | Retrieval | test | (text, audio) | R@1 |
| Clotho (Drossos et al., 2020) | | test | | |
| VGGSound (Chen et al., 2020) | Retrieval | test | (visual, audio) | R@1 |
| MusicCaps (Agostinelli et al., 2023) | | test | | |
| MMAU (Sakshi et al., 2025) | Audio QA | test-mini | (audio, text) | Acc% |
| MMAR (Ma et al., 2025) | | test | | |

# 5 EXPERIMENTAL RESULTS

## 5.1 OVERALL RESULTS

The results of our model are shown in Table 3 and Table 4. WAVE demonstrates its capabilities to generate versatile multimodal embeddings, achieving strong performance across a range of video, audio, and audio-visual scenarios. In the video domain, WAVE comprehensively outperforms existing open-source models across all sub-tasks on the video track of MMEB-v2, which systematically evaluates various video understanding capabilities. Notably, the overall performance of our model even surpasses that of the industrial-grade model, Seed-1.6-Embedding [1]. Furthermore, our model also exhibits strong performance on LoVR, not only on caption-based text-to-clip retrieval but also on concept-based theme-to-clip retrieval, leading existing open-source multimodal embedding LLMs.

Table 3: Results of video embedding benchmarks. Models are evaluated on the video track of MMEB-v2 and LoVR.

| Model | MMEB-v2-Video | | | | | LoVR | |
|---|---|---|---|---|---|---|---|
| | Overall | CLS | QA | RET | MRET | text-to-clip | theme-to-clip |
| LamRA 7B (Liu et al., 2025a) | 35.0 | 39.3 | 42.6 | 24.3 | 32.8 | 62.9 | 60.2 |
| GME 7B (Zhang et al., 2024) | 38.4 | 37.4 | 50.4 | 28.4 | 37.0 | 51.2 | 43.9 |
| CAFe 7B (Yu et al., 2025a) | 42.4 | 35.8 | 58.7 | 34.4 | 39.5 | - | - |
| Seed-1.6-Embedding | 55.3 | 55.0 | 60.9 | 51.3 | **53.5** | - | - |
| WAVE 7B | **59.9** | **57.8** | **72.5** | **54.7** | 50.8 | **62.9** | **66.0** |

Table 4: Results of audio and audio-visual embedding benchmarks. Different tasks are evaluated, including audio retrieval (A-RET), audio-visual retrieval (AV-RET) and audio QA (A-QA).

| Method | A-RET | | AV-RET | | A-QA | |
|---|---|---|---|---|---|---|
| | AudioCaps | Clotho | VGGSound | MusicCaps | MMAU | MMAR |
| Reference Model | (Mei et al., 2024) | | encoder-only retrieval model (ours) | | Qwen2.5-Omni 7B | |
| Reference Value | 42.2 | 21.5 | 10.3 | 8.6 | 71.5 | 56.7 |
| WAVE 7B | **44.2** | **25.6** | **25.0** | **20.4** | **76.6** | **68.1** |

In the audio domain, on the widely used AudioCaps and Clotho datasets, WAVE achieves superior audio retrieval performance compared to previous models that rely on separate-encoder architectures. Moreover, as a unified multimodal embedding model, WAVE is also capable of video-to-audio retrieval, a more challenging task that directly bypasses the text modality. For a fair comparison, we

---

[1]https://seed.bytedance.com/en/blog/built-on-seed1-6-flash-seed-1-6-embedding-launched

train an encoder-only retrieval model (columns 4 and 5 in Table 4) using the same video-to-audio retrieval data, where video embeddings are extracted by WAVE's visual encoder and audio embeddings are extracted by WAVE's speech and audio encoders. The results in Table 4 show that WAVE considerably outperforms the encoder-only retrieval model on audio-visual retrieval, not only on the in-domain VGGSound test set but also on the out-of-domain video-to-music MusicCaps data.

More evaluation results are shown in Appendix C.

## 5.2 ANALYSIS OF PROMPT-AWARE EMBEDDINGS

Beyond retrieval, WAVE leverages its LLM backbone's reasoning to produce prompt-aware embeddings conditioned on textual instructions (see Appendix D for a case study). The ability to follow instructions is crucial for embedding MLLMs in QA tasks, which can be shown by the following question example from Video-MME (Fu et al., 2025): *Which of the following features/items is not discussed in the video in relation to the tomb? A. Inkstone. B. Niche. C. Jade. D. Sacrificial table.*

Models that cannot understand the input question may probably produce an embedding that represents the main content of the video, which will lead to incorrect predictions. We list the detailed QA results in Table 5. To investigate the extent to which text prompts contribute to WAVE's ability to generate embeddings, we also instructed WAVE with a common prompt ("Please describe the video") when testing, instead of separate questions.

Table 5: Results of different models on MMEB-v2 video QA data, including Video-MME (Fu et al., 2025), MVBench (Li et al., 2024), NExT-QA (Xiao et al., 2021), EgoSchema (Mangalam et al., 2023), and ActivityNetQA (Yu et al., 2019). In the case of "w/ separate questions", each question is used as a different prompt.

| Model | MMEB-v2-Video QA | | | | | |
|---|---|---|---|---|---|---|
| | Average | Video-MME | MVBench | NExT-QA | EgoSchema | ActivityNetQA |
| LamRA 7B | 42.6 | 34.1 | 37.2 | 43.7 | 44.8 | 53.2 |
| GME 7B | 50.4 | 39.2 | 46.6 | 53.6 | 46.8 | 65.6 |
| CAFe 7B | 58.7 | 46.0 | 48.9 | 62.4 | 60.0 | 76.0 |
| Seed-1.6-Embedding | 60.9 | 54.0 | 53.3 | 66.2 | 52.2 | 78.6 |
| WAVE 7B, w/ a common prompt | 51.8 | 39.3 | 44.7 | 53.5 | 61.4 | 60.2 |
| WAVE 7B, w/ separate questions | **72.5** | **63.4** | **69.6** | **82.6** | **66.2** | **80.9** |

Compared with existing embedding MLLMs, WAVE outperforms strong baselines when testing with separate questions, averaging about 12% higher than Seed-1.6-Embedding. However, using general prompts to extract embeddings leads to a drastic performance degradation across all QA datasets. This stark contrast not only highlights the strong instruction-following capability of WAVE, but also suggests the critical limitation of a single, static representation for complex tasks like multimodal QA.

On audio-reasoning benchmarks, WAVE further surpasses its base, Qwen2.5-Omni model, as Table 4 shows. This is notable given that WAVE was trained only to generate question-conditioned embeddings for video QA. This cross-modal transfer underscores robust generalisation and supports the hypothesis that WAVE learns a unified, modality-agnostic embedding space.

## 5.3 BENEFIT OF JOINT MULTI-MODAL, MULTI-TASK TRAINING

A core hypothesis behind our unified model is that joint training across diverse modalities and tasks fosters a more robust and powerful universal embedding space. We posit that learning from audio, video, and text data simultaneously enables positive knowledge transfer, where insights from one modality can enhance the understanding of another. To verify this, we conducted an ablation study comparing our fully-trained WAVE model against specialist models trained on modality-specific subsets of the data.

Specifically, using the data described in Table 1, we train models under the following three task settings: training video-text retrieval and video-QA, training audio-text retrieval, and training video-audio retrieval. Each model is trained on only one fixed pair of modalities, without mixing data from other modalities. Then we test the three separately trained models on video, audio, and audio-visual

benchmarks, respectively. The results are shown in Table 6, denoted as "Separate". The final WAVE model jointly trained across all modalities is also reported for comparison, denoted as "Joint".

Table 6: Comparison of model performance under separate vs. joint training schemes. The model jointly trained on all modalities and tasks consistently outperforms specialist models trained on separate modality-task pairs.

| Training | MMEB-v2-Video | | | | | A-RET | | AV-RET | |
|---|---|---|---|---|---|---|---|---|---|
| | Overall | CLS | QA | RET | MRET | AudioCaps | Clotho | VGGSound | MusicCaps |
| Separate | 58.2 | 57.5 | 71.6 | **56.1** | 47.6 | 42.5 | 24.0 | 24.9 | 20.1 |
| Joint | **59.0** | **57.8** | **72.5** | 54.7 | **50.8** | **44.2** | **25.6** | **25.0** | **20.4** |

As shown in Table 6, the model trained jointly across modalities outperforms separately trained specialist models on seven of eight tasks, indicating positive cross-modal knowledge transfer. Exposure to richer, more diverse signals encourages learning generalised, modality-agnostic semantic representations rather than modality-specific features, underscoring the promise of a model for general-purpose embedding extraction.

## 5.4 ANALYSIS OF FEATURE FUSION

When using an LLM for embedding extraction, a common choice is last-token pooling, which takes the EOS token's hidden state from the final layer as the sequence representation. However, as Gou et al. (2025) observe, different LLM layers specialise in distinct functions for video understanding, implying complementary information is distributed across depth. Accordingly, we aggregate signals from all layers to form the final embedding, preserving both low-level perceptual cues and high-level semantic reasoning. Concretely, we collect the last-token states from every layer, concatenate them, and feed them to a lightweight fusion module to produce the output embedding.

To efficiently assess embedding-extraction strategies while conserving compute, we conduct an expanded ablation primarily on pure-visual (no-audio) video retrieval, with an additional check in the audio-visual setting. Evaluations use the MMEB-v2 video-retrieval split. Beyond the two main strategies, **1)** standard last-token pooling from the final LLM layer and **2)** our all-layer last-token MLP fusion, we also test **3)** the last-token output from the first layer, **4)** the last-token from a middle layer (Layer 15), and **5)** a learnable weighted sum (Peters et al., 2018) of last-token features across all layers. The LLM has twenty-eight layers in total. Results are shown in Table 7.

Table 7: Results of embedding extraction methods on the MMEB-v2 video retrieval data, including MSR-VTT, VATEX, MSVD, DiDeMo, and YouCook2. Note that videos in MSR-VTT, VATEX, and YouCook2 are paired with audio. "V" and "A+V" refer to visual-only and audio-visual, respectively.

| Method | Modality | MMEB-v2-Video RET | | | | | |
|---|---|---|---|---|---|---|---|
| | | Average | MSR-VTT | VATEX | MSVD | DiDeMo | YouCook2 |
| Last token pooling (first layer) | | 38.8 | 44.8 | 37.3 | 60.9 | 39.0 | 12.0 |
| Last token pooling (middle layer) | | 45.0 | 48.9 | 41.4 | 67.5 | 47.3 | 17.8 |
| Last token pooling (last layer) | V | 49.6 | 52.1 | 46.2 | **69.7** | 53.0 | 27.2 |
| All-layer last token weighted sum | | 48.3 | 49.4 | 45.6 | 69.4 | 50.6 | 26.3 |
| All-layer last token MLP fusion | | **50.5** | **53.6** | **47.5** | 68.7 | **55.4** | **27.3** |
| Last token pooling (last layer) | A+V | 54.7 | 58.2 | 56.3 | 69.3 | 54.8 | 34.9 |
| All-layer last token MLP fusion | | **56.1** | **58.5** | **58.4** | 69.3 | **57.4** | **36.8** |

As shown in Table 7, using only first-layer or middle-layer features yields a marked drop versus the final-layer representation, consistent with a hierarchical abstraction in which the top layer carries the most semantically relevant information for retrieval. Early-layer cues are still useful, however: fusing last-token features from all layers with a small MLP consistently surpasses the strong last-layer baseline. By contrast, a direct weighted sum across layers underperforms, suggesting that cross-layer interactions for video tasks are complex and non-linear, and thus benefit from learned transformations. The pattern holds in the audio-visual setting, our all-layer MLP fusion again out-

performs last-layer pooling, and Table 7 further shows that audio substantially boosts video retrieval, reinforcing the value of a unified, general-purpose multi-modal embedding model.

## 6 CONCLUSION

We present WAVE, to our knowledge, the first unified, versatile audio–visual embedding MLLM that maps text, audio, silent video, and synchronised audio–visual inputs into a single semantic space. A dual audio-encoder design combined with hierarchical all-layer feature fusion yields robust, comprehensive multimodal representations. Joint multi-modal, multi-task training enables WAVE to achieve strong results (e.g., on the MMEB-v2 video track) and to generate prompt-aware embeddings that translate into competitive multimodal QA performance. Ablations confirm the benefits of unification—showing positive cross-modal transfer and the value of learned cross-layer fusion. WAVE establishes a new, powerful baseline for universal audio-visual representation learning and can serve as a springboard for cross-modal, any-to-any applications.

## 7 REPRODUCIBILITY STATEMENT

We provide detailed descriptions of the model architecture, training pipelines, training data and hyperparameters in Sections 3.1, 3.2, 4.1, and 4.2. All datasets, code, and model checkpoints will be released. These provide enough reproducibility for our work.

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

## A  THE USE OF LARGE LANGUAGE MODELS

We used Gemini-2.5-Pro to help us check for grammar errors and polish the fluency of our sentences.

## B  INFERENCE PROCEDURE FOR EVALUATION TASKS

The evaluation tasks can also be divided into two categories: retrieval tasks and QA tasks. Tasks that use a single, unified general prompt to extract embeddings for all test samples are regarded as retrieval tasks. This includes the classification, retrieval, and moment retrieval subsets of MMEB-v2-Video. Conversely, tasks that use separate and specific questions as prompts to extract embeddings are considered QA tasks.

For all retrieval tasks, we generate an embedding for each sample in the test set based on its input video/audio, using the fixed text prompt, "Please describe the video/audio." If the task involves retrieving text, the corresponding ground-truth text captions for each sample are also used to generate text embeddings. The entire test set then forms the candidate pool for the retrieval task. The candidate with the highest similarity score to the query embedding is selected as the model's prediction.

For QA tasks, all QA evaluations for the embedding LLMs are performed using an embedding-based methodology. To be specific, the inference procedure is as follows:

1. The model first generates a single embedding that is conditioned on both the source modality (video/audio) and the provided question text.
2. Separately, the model generates an embedding for the text of each answer option.
3. The similarity between the question-conditioned video/audio embedding and each of the option text embeddings is then calculated.
4. The option with the highest similarity score is selected as the model's predicted answer.

## C  MORE EVALUATION RESULTS OF WAVE

In the retrieval task of MMEB-v2-Video, only one direction, text-to-video retrieval, was evaluated. Similarly, for the audio retrieval and audio-visual retrieval tasks presented in Table 4, only a single direction was assessed, i.e, text-to-audio and video-to-audio retrieval, respectively. This focus was chosen because these directions more closely align with practical, real-world application scenarios like multimodal search and recommendation. However, it is undeniable that performance in the reverse direction is also important. In Table 8, we provide supplementary results for video retrieval (V-RET), audio retrieval (A-RET), and audio-visual retrieval (AV-RET) in the other direction. WAVE can also achieve competitive results in the other direction.

## D  CASE STUDY OF PROMPT-AWARE EMBEDDINGS

To provide a more intuitive and qualitative demonstration of WAVE's prompt-aware embedding capability, we conduct a case study on a single video. We select a video from the MSR-VTT test set,

Table 8: Results of video-to-text, audio-to-text and audio-to-video retrieval. Corresponding reference models and their scores are also provided.

| Method | V-RET | | | A-RET | AV-RET |
|---|---|---|---|---|---|
| | MSR-VTT | VATEX | YouCook2 | Clotho | VGGSound |
| Reference Model | GME 7B (Zhang et al., 2024) | | | (Mei et al., 2024) | encoder-only retrieval model (ours) |
| Reference Value | 32.2 | 33.0 | 10.0 | **27.1** | 8.2 |
| WAVE 7B | **55.1** | **55.4** | **29.8** | 24.5 | **25.8** |

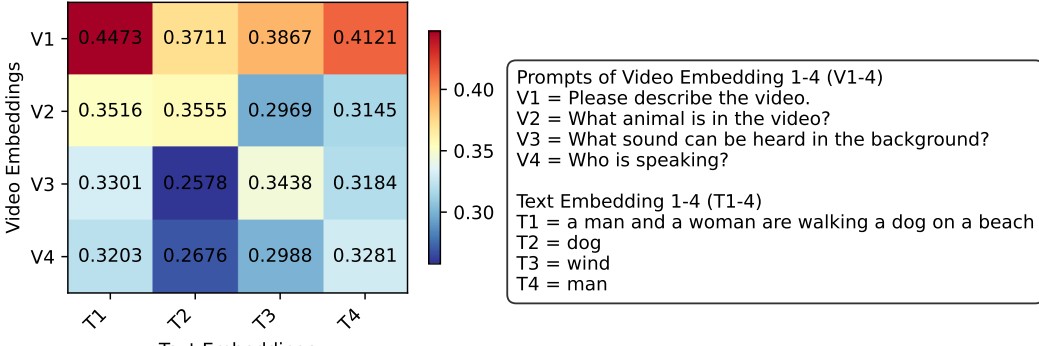

Figure 2: A heatmap visualizing the cosine similarity between video embeddings (V1-V4) and text embeddings (T1-T4). All four video embeddings are generated from the **same video** but conditioned on different textual prompts. The text embeddings represent various concepts present in the video.

which shows a man and a woman walking a dog on a beach, with wind blowing in the background. We then generated four distinct embeddings (V1-V4) for this single video conditioned on different prompts, ranging from a general description request to specific questions about visual animals, background sounds, or speakers. Concurrently, we generated four text embeddings (T1-T4) representing a general description and the specific concepts of "dog", "wind", and "man".

The cosine similarities between these video and text embeddings are visualised in the heatmap in Figure 2. The general prompt V1 ("Please describe the video.") yields an embedding that has the highest similarity (0.4473) with the general text description T1. In addition, video embeddings conditioned on prompts about specific aspects of the video (V2, V3, V4) also show high similarity to T1, indicating that these prompt-aware embeddings still retain the overall semantic context of the video. In addition, video embeddings generated from specific prompts are clearly biased towards the textual representation of that specific feature. For instance, the embedding V2, generated by the prompt "What animal is in the video?", has a slightly higher similarity than T1 and a significantly higher similarity with T2 ("dog") than with T3 ("wind") or T4 ("man"). Similarly, the audio-focused V3 aligns best with "wind" (T3), and the speaker-focused V4 matches most closely with "man" (T4). This clearly demonstrates that WAVE can dynamically shift the semantic focus of its output embedding to produce a representation that is precisely tailored to the user's query.

# E  ANALYSIS OF DUAL SPEECH & AUDIO ENCODERS

Speech and general audio events are both crucial elements within an audio signal. Our base model, Qwen2.5-Omni, possesses some capability to process general audio, but its encoder for audio processing is derived from Whisper (Radford et al., 2023), a model optimised for automatic speech recognition. This speech encoder, which is primarily specialised for modelling speech, has an insufficient understanding of non-speech audio events. To address this limitation, we augment the existing speech encoder with a dedicated audio encoder, BEATs (Chen et al., 2022b), which is designed for comprehensive audio event understanding.

To validate the effectiveness of this dual-encoder approach, we compared its performance on video retrieval, audio retrieval, and audio-visual retrieval, respectively, with that of using only the original speech encoder. For video retrieval, we test the models on MSR-VTT, VATEX, and YouCook2, whose videos are paired with audio. The results are presented in Table 9. The dual-encoder configuration consistently outperforms the single speech encoder on both the audio retrieval and audio-visual retrieval, and achieves comparable or better performance on video retrieval benchmarks. This indicates that the speech and audio encoders can complement each other to further enhance the model's ability to interpret both speech and environmental sounds.

Table 9: Results of using dual speech and audio encoders and using a speech encoder only. Video retrieval (V-RET), audio retrieval (A-RET), and audio-visual retrieval (AV-RET) are evaluated here.

| Method | V-RET | | | A-RET | | AV-RET | |
|---|---|---|---|---|---|---|---|
| | MSR-VTT | VATEX | YouCook2 | AudioCaps | Clotho | VGGSound | MusicCaps |
| Single speech encoder | **58.6** | 56.6 | 34.3 | 39.6 | 22.4 | 23.3 | 18.3 |
| Dual speech & audio encoders | 58.5 | **58.4** | **36.8** | **42.5** | **24.0** | **24.9** | **20.1** |

## F  THE EFFECT OF IMAGE TRAINING

WAVE is primarily trained on video data and achieves strong performance on video retrieval-related tasks. However, this success is not due to a narrow specialization in video. In fact, WAVE is not specialized only for video retrieval, and image data actually helps. In a preliminary study, we compared the results of training solely on video data against training on a mixture of video data and a nearly equal amount of image data from the MMEB-v1 training set. This experiment utilized the same pure-visual video retrieval setup as described in Section 5.4. Table 10 presents the results of this comparison.

Table 10: Comparison of results for training with and without image data. We evaluate text-to-image retrieval (Liu et al., 2021) on the VisualNews dataset and text-to-video retrieval on MMEB-v2-Video.

| Training Data | Image RET | MMEB-v2-Video RET | | | | | |
|---|---|---|---|---|---|---|---|
| | VisualNews | Average | MSR-VTT | VATEX | MSVD | DiDeMo | YouCook2 |
| Video Data | 54.9 | 49.6 | 52.1 | 46.2 | **69.7** | 53.0 | 27.2 |
| Video Data + Image Data (MMEB-v1) | **74.8** | **50.2** | **52.4** | **46.2** | 69.0 | **55.8** | **27.8** |

As the results show, including image data substantially improves image retrieval and provides a slight gain in video retrieval. This shows that WAVE's strong performance is not due to avoiding image tasks or specializing exclusively in video.

