# OpenReview forum: "WAVE: Learning Unified & Versatile Audio-Visual Embeddings with Multimodal LLM"
_ICLR.cc/2026/Conference — ICLR 2026 Oral_

### Official Review · Reviewer_gqQX · 2025-10-30

**Soundness:** 3
**Presentation:** 3
**Contribution:** 3
**Rating:** 8
**Confidence:** 4

**Summary:**

This paper proposes a method to fine-tune a multimodal large language model (MLLM) to produce prompt-aware embeddings for audio and video. Starting from Qwen2.5-Omni, the authors first learn a BEATs-based adapter to ingest an additional audio input. They then apply contrastive training with LoRA using two objectives: bidirectional cross-modal retrieval and a QA objective that treats the source modality to text as retrieval. Text embeddings are taken from the last token of the last layer. For non-text modalities, the method conditions on a prompt, aggregates the last tokens from all layers, and passes them through a fusion module. Experiments show improved retrieval performance over other MLLMs, and the fine-tuning also improves the base MLLM’s text generation task performance.

**Strengths:**

The task formulation and model design are clear and sensible. The experiments are comprehensive and the results strongly support the efficiency of the proposed method. The presentation is clear.

**Weaknesses:**

In the evaluation benchmarks, retrieval tasks appear to be reported in a single direction only, for example text-to-audio on Clotho. Reporting both directions would provide a fuller picture.

The use of the term “QA” is potentially confusing. During training, the QA objective is a source-modality-to-text contrastive setup, while in evaluation some QA tasks are executed via text generation with the MLLM. The same ambiguity appears for other evaluation benchmarks. It would help to specify, for each evaluation task, the exact inference procedure.

**Questions:**

1. What is the architecture of the BEATs aligner, and is it similar to the fusion module?
2. How do text prompts vary across training tasks and evaluation tasks?

---

> ### Author Response · Authors · 2025-11-19
>
> We are very grateful to the reviewer for the insightful comments and constructive suggestions, which have been very helpful for improving our work. In response to the feedback from all three reviewers, we have revised the paper, and the changes are indicated in blue text within the manuscript. In the following, we provide a point-by-point response to your concerns.
>
> ---
> **Weakness 1: Regarding the evaluation of retrieval tasks in the reverse direction.**
>
> We focused on the text-as-query direction (e.g., text-to-video/audio) in the main paper because it is the most common setting in real-world applications and the direction for which WAVE is primarily optimised. However, we agree that reporting the reverse directions provides a more complete evaluation.
>
> To address this, we now include results for video-to-text (v2t), audio-to-text (a2t), and audio-to-video retrieval (a2v). As shown below, WAVE also performs strongly in these directions:
>
> | | MMEB-v2-Video (v2t) | Clotho (a2t) | VggSound (a2v) |
> | :--- | :--- | :--- | :--- |
> | Reference Model | GME 7B | CNN14-BERT-FT [1] | Encoder-only retrieval model (ours) |
> | Reference Value | 32.3 | 27.1 | 8.2 |
> | WAVE | 52.3 | 24.5 | 25.8 |
>
> We have added these results in **Appendix C (Line 760)** in our revised manuscript.
>
> [1] WavCaps: A ChatGPT-assisted Weakly-labelled Audio Captioning Dataset for Audio-language Multimodal Research, Mei et al, IEEE/ACM Transactions on Audio, Speech, and Language Processing, 2024
>
> ---
> **Weakness 2: Regarding the inference procedure for QA tasks.**
>
> In fact, in this paper, all QA evaluations for the embedding LLMs are performed using an embedding-based methodology. To be specific, the inference procedure is as follows:
>
> 1. The model first generates a single embedding that is conditioned on both the source modality (video/audio) and the provided question text.
> 2. Separately, the model generates an embedding for the text of each answer option.
> 3. The similarity between the question-conditioned video/audio embedding and each of the option text embeddings is then calculated.
> 4. The option with the highest similarity score is selected as the model's predicted answer.
>
> We have added these explanations in **Appendix B (Line 739)** in our updated manuscript to ensure the evaluation procedure is clear.
>
> ---
> **Question1: What is the architecture of the BEATs aligner, and is it similar to the fusion module?**
>
> The BEATs aligner is a 2-layer MLP with GELU, as Section 4.1 mentions (Line 252), which is indeed the same as the fusion module.
>
> ---
> **Question 2: How do text prompts vary across training tasks and evaluation tasks?**
>
>
> We categorise tasks into two prompt types: retrieval and QA.
>
> - **Retrieval tasks**: Both during training and evaluation, we use **a single, unified general prompt** to extract embeddings for all samples. The prompt is: “Please describe the video.” This keeps the embedding task-consistent and computationally efficient.
> - **QA tasks**: The text prompt is the question itself, which varies from sample to sample and can target any specific aspect of the video/audio. These question-conditioned prompts enable the model to extract embeddings aligned with the query semantics.
>
> This distinction is applied consistently in all training and evaluation settings.

---

> > ### Comment · Reviewer_gqQX · 2025-11-26
> >
> > Thanks for the detailed explanation. The replies address my concerns and questions.

---

### Official Review · Reviewer_mAPG · 2025-10-31

**Soundness:** 3
**Presentation:** 3
**Contribution:** 2
**Rating:** 6
**Confidence:** 4

**Summary:**

This paper introduces WAVE, a LLM-based embedding model specifically designed to create a unified representation space for text, audio, silent video, and synchronized audio-visual inputs.

To achieve this versatility, WAVE employs a hierarchical feature fusion strategy that aggregates representations from multiple LLM layers, alongside a dual-encoder architecture for audio inputs.

The model is optimized through a joint multi-modal, multi-task training approach to enable any-to-any cross-modal retrieval and the generation of prompt-aware embeddings that condition on user instructions for tasks like multimodal QA.

Experimentally, WAVE outperforms other baselines on the MMEB-v2 video benchmark and yields descent results in audio and video-to-audio retrieval, with ablation studies confirming the performance benefits derived from both joint training and the learned cross-layer fusion technique.

**Strengths:**

* The writing is easy to follow.

*  The architecture features an effective hierarchical feature fusion strategy by aggregating representations across multiple LLM layers and a dual-encoder design for audio. Ablation studies confirm that the proposed joint multi-modal, multi-task training strategy enables positive cross-modal knowledge transfer and superior results compared to specialist models.

*  WAVE achieves new state-of-the-art results on the MMEB-v2 video benchmark and shows superior performance in audio and video-to-audio retrieval compared to strong baselines.

**Weaknesses:**

1.  A major limitation is the model's reliance on prompt-aware embeddings for high performance in complex tasks like multimodal QA. Using a single general prompt to extract the embedding results in a drastic performance degradation across all QA benchmarks, highlighting the critical limitation of a single, static representation in handling complex query semantics. Although boosting performance, generating diverse embeddings can be very expensive in real-world scenarios. Consider the model that needs to generate different embeddings based on various user inputs.


2.  Achieving optimal performance requires a complex, learned MLP fusion across the last-token outputs of all LLM layers. Ablation studies showed that simpler aggregation methods, such as a direct weighted sum across layers, underperform, suggesting the necessary cross-layer interactions are highly complex. This leaves readers wondering whether all the layers were necessary, or if there are other strategies to select meaningful layers that would be sufficient.

3.   As an LLM-based model built on a 7B parameter backbone, Qwen2.5-Omni, the training demands are significant, requiring large-scale infrastructure (192 H20 GPUs for approximately 36 hours), which is hard for many labs to reproduce.

**Questions:**

### 1. Analysis of Prompt-Aware Embeddings and Generalization

*  Could the authors elaborate on the fundamental limitations preventing a single, static embedding from adequately capturing complex query semantics for QA? Does this performance gap imply that for high-level reasoning tasks, WAVE’s LLM backbone is using the prompt to select relevant features internally rather than deriving a truly universal, task-agnostic representation?

*  For users who need a generalized embedding for downstream applications that lack explicit questions, what is the optimal and computationally lightest prompt recommended by the authors that can minimize performance loss while maintaining acceptable semantic coverage?

### 2. Feature Fusion and Interpretability

* Given that a direct weighted sum underperforms, suggesting that cross-layer interactions are complex and non-linear, can the authors provide deeper insights into what the two-layer MLP fusion module learns? Are there any visualization techniques or analysis that can illustrate the relative importance assigned to low-level (early-layer) perceptual cues versus high-level (late-layer) semantic reasoning during the fusion process?

### 3. Dual-Encoder Necessity and Redundancy

* Since BEATs is designed for comprehensive audio event understanding, did the authors explore replacing the existing speech encoder entirely with a second, possibly smaller, instance of the BEATs encoder, or fine-tuning a single, unified audio encoder? If the dual approach is mandated by specialized roles, how do the embeddings from the two encoders contribute uniquely to the final unified representation, beyond the observed performance boost in audio retrieval?

### 4. Inference and Computational Cost

*  While the training resources are impressive (192 H20 GPUs for 36 hours), could the authors provide a comparison of the average latency and total inference computation (FLOPs or time per sample) required to generate a WAVE embedding versus competing MLLM-based embedding models (e.g., LamRA or CAFe)? Specifically, how much overhead is introduced by processing and fusing features from all layers compared to standard last-token pooling from only the final layer?

---

> ### Author Response · Authors · 2025-11-19
>
> We are very grateful to the reviewer for the insightful comments and constructive suggestions, which have been very helpful for improving our work. In response to the feedback from all three reviewers, we have revised the paper, and the changes are indicated in blue text within the manuscript. In the following, we provide a point-by-point response to your concerns.
>
> ---
> **Weakness 1 & Question 1: Regarding the performance of prompt-aware embeddings versus static embeddings.**
>
> **Regarding the suggested limitations of a single, static embedding for QA**: The performance degradation of a single static embedding on QA benchmarks is largely due to the wide diversity of question types. A static embedding is naturally tied to the dominant content of the video and therefore struggles to capture complex or atypical query semantics. For example, VideoMME includes negative reasoning questions such as "Which feature is not discussed in the video?", which requires the model to reason beyond what is present in the core representation. Similarly, for highly specific queries, e.g., "How many guitar-shaped instruments are in the video?" where the answer is simply "7", a static, content-focused embedding cannot easily be aligned with such a concise answer without question guidance.
>
> In contrast, WAVE’s prompt-conditioned embeddings explicitly incorporate question semantics, enabling the LLM backbone to highlight and prioritize the relevant features. The observed performance gap therefore indicates that, for high-level reasoning tasks, WAVE relies on prompt-guided feature selection rather than a single, universal, task-agnostic embedding.
>
> **Regarding the optimal prompt for downstream applications**: For downstream applications without explicit questions (e.g. recommendation, retrieval, or semantic search), the concerns above are far less pronounced. In these use cases:
>
> - User queries generally relate to the main content of the video rather than negation-based or adversarial reasoning.
> - Queries tend to be more descriptive (e.g., “video with 7 guitar-shaped instruments”), avoiding the mismatch seen when answers are extremely short.
> - As shown on EgoSchema (Table 5), which focuses on global video understanding, a single unified prompt performs very well.
>
> Therefore, **for such scenarios, we recommend to use our standard retrieval prompt to achieve both computationally efficient and semantically robust**. It provides a strong, general-purpose embedding with minimal performance loss while maintaining broad semantic coverage.
>
> ---
> **Weakness 2 & Question 2: Regarding feature fusion and interpretability**
>
> To gain deeper insights into what the two-layer MLP fusion module learns, we first conducted an experiment where we selectively zeroed out the features from different layers at the MLP's input during inference. Unfortunately, this experiment revealed that zeroing out the features from **any single layer** caused the resulting embeddings to become meaningless (The top-1 recall dropped to less than 0.1%, and even the top-20 recall remained below 2%). This strongly suggests that the two-layer MLP has learned a complex transformation that utilizes information from every single layer, rather than ignoring any of them. This, in turn, indicates that the information from each layer is meaningful to the fusion process.
>
> However, from an empirical standpoint, we still hypothesized that the late-layer features hold greater importance. Building on the experimental setup from Table 7, we tested this by fusing the last layer's features with either the middle or first layer's features, using an MLP to extract the final embedding. The results are presented in the table below, with some results from Table 7 included for easier comparison.
>
> | Method | MMEB-v2-Video RET |
> | :--- | :--- |
> | Last token pooling (first layer) | 38.8 |
> | Last token pooling (middle layer) | 45.0 |
> | Last token pooling (last layer) | 49.6 |
> | 2-layer last token MLP fusion (last & first layers) | 49.9 |
> | 2-layer last token MLP fusion (last & middle layers) | 49.6 |
> | All-layer last token MLP fusion | 50.5 |
>
> The results show that using only the middle or first layer features leads to a significant performance drop compared to using only the last layer, confirming that **the last layer's features play the most critical role**.
>
> However, when we fuse the last layer's features with those from the middle or first layers, the performance shows a slight improvement over using the last layer alone. This suggests that the earlier layers provide **auxiliary features that enhance the model's robustness**. This enhancement becomes even more pronounced in the all-layer fusion setting, where the model's retrieval performance is further improved.

---

> > ### Author Response · Authors · 2025-11-19
> >
> > **Weakness 3 & Question 4: About inference and computational cost**
> >
> > **Regarding the training demands and reproducibility**: WAVE is intended as a foundational multimodal embedding model, and we will release both the code and pretrained weights. This enables labs with limited compute to fine-tune WAVE on their own tasks without needing to reproduce the full pre-training run.
> >
> > **Regarding inference latency and computational overhead**: Although training is resource-intensive, inference overhead is modest. We benchmarked latency on MSR-VTT (1,000 videos, ~15 seconds each at 2 FPS), comparing LamRA, our last-layer pooling variant, and our full all-layer MLP fusion model. The results are summarised below:
> >
> > | Model | Time per sample / second |
> > | :--- | :--- |
> > | LamRA 7B | 0.410 |
> > | Ours, Last token pooling (last layer) | 0.508 |
> > | Ours, All-layer last token MLP fusion | 0.519 |
> >
> > Compared to LamRA, our model is roughly 20% slower, **mainly because WAVE processes both audio and visual streams**, while LamRA is vision-only. Importantly, the overhead introduced specifically by all-layer fusion is very small. Relative to our own last-layer pooling variant, the full MLP fusion **adds only 0.011 seconds per sample (~2%)**. This is because the dominant cost lies in the LLM backbone forward pass, whereas the two-layer fusion module is computationally lightweight. As a result, WAVE’s inference speed remains very close to standard last-token pooling, despite leveraging richer cross-layer features.
> >
> > ---
> > **Question 3: About dual-encoder necessity and redundancy**
> >
> > We did not replace the native speech encoder with BEATs. WAVE is built on Qwen2.5-Omni, whose LLM backbone is tightly aligned with its original speech encoder; removing it would likely harm overall performance. Moreover, to our knowledge, **no single unified audio encoder currently achieves state-of-the-art performance on both speech and general audio events**, as existing encoders tend to specialize in one domain.
> >
> > To understand the distinct contributions of each encoder, we compared using only the Qwen2.5-Omni speech encoder versus using our dual-encoder setup. The results are in Appendix E (Table 9, Line 838). The findings are:
> >
> > - The native speech encoder already provides strong performance across tasks, indicating it remains the dominant contributor.
> > - Adding BEATs yields a consistent, modest improvement, suggesting it introduces **complementary information** about general audio events that the speech encoder alone does not capture.
> >
> > Together, the two encoders provide a more robust and semantically richer representation than either could achieve individually, supporting the design choice of dual-encoder architecture.

---

> > > ### Author Response · Authors · 2025-11-26
> > >
> > > Dear Reviewer mAPG,
> > >
> > > We hope that our responses have adequately addressed the concerns you raised in your review, and we would greatly appreciate your further feedback. Please let us know if there is anything further we can clarify.
> > >
> > > Best regards, The Authors

---

### Official Review · Reviewer_zEZy · 2025-11-01

**Soundness:** 3
**Presentation:** 3
**Contribution:** 2
**Rating:** 4
**Confidence:** 4

**Summary:**

The paper introduces Wave, a unified multimodal embedding model capable of embedding text, audio, video and audio-video inputs. Wave is validated on a variety of video, audio and text retrieval tasks. Furthermore, Wave is capable of robustly using instructions to improve multimodal embedding performence.

**Strengths:**

- Introducing audio to multimodal retrieval is relatively novel. The performance of the model is quite strong.
- Using a concatenation of many layers last token hidden is a novel method for vector embedding in this setting.
- The manuscript is clear and well-written.

**Weaknesses:**

- The approach is very similar to [1] except an omni-model is used instead of a VLM.

- The paper is missing implementation details relating to model architecture, including how inputs are templated into LLM backbone, resolution and visual tokenization in the vision encoder.

- The paper does not compare their proposed pooling strategy with mean pooling with bidirectional attention, which tends to outperform last-token pooling strategies [2].

- The evaluations of audio-visual retrieval is limited. The reviewer can only find a single task that tests WAVE’s ability to retrieve audio-visual items (RET split of MMEB-v2-Video), and only in one direction. Furthermore, It is unclear if MMEB-v2-Video is designed to have audio used its retrieval task. This could result in the audio track trivializing the task or being useless.

- It is unclear whether WAVE’s superior video retrieval performance comes from its different techniques or larger scale video retrieval training or its specialization into video retrieval (I.e. no image retrieval).

Minor:
- Potentially missing related work: the use of “distractor” answers for QA training seems very similar to [3].

**Questions:**

Several question can be found in the above weaknesses section, in addition:
- How were the “distractor“ answers used during QA training created?

---

> ### Author Response · Authors · 2025-11-19
>
> We are very grateful to the reviewer for the insightful comments and constructive suggestions, which have been very helpful for improving our work. In response to the feedback from all three reviewers, we have revised the paper, and the changes are indicated in blue text within the manuscript. In the following, we provide a point-by-point response to your concerns.
>
> ---
> **Weakness 1: The approach is very similar to [1] except an omni-model is used instead of a VLM.**
>
> Thank you for pointing this out. The specific citation for paper [1] was not provided in the review, so we are unsure which prior work the comment refers to. We would greatly appreciate it if the reviewer could provide the full reference. This will allow us to carefully examine the work and clearly articulate the similarities and differences between it and WAVE in the revised manuscript.
>
> ---
> **Weakness 2: The paper is missing implementation details relating to model architecture, including how inputs are templated into LLM backbone, resolution and visual tokenization in the vision encoder.**
>
> Thank you for the comment. We have added the missing implementation details in **Section 3.1 (Line 168) and Section 4.1 (Line 261)** of the revised manuscript. A brief summary is provided below:
>
> 1. **Auditory tokenization**. For waveform input, the speech encoder and audio-event encoder produce token sequences with identical lengths because they operate on the same input and share the same time–frequency resolution. We therefore interleave speech tokens and audio-event tokens one-to-one to form a unified auditory token sequence.
> 2. **Audio–visual tokenization**. For multimodal input, we divide both auditory and visual token sequences into N temporal segments (corresponding to sampled frames) and interleave them segment-by-segment, ensuring consistent temporal alignment.
> 3. **Vision encoder and resolution**. Video frames are processed at up to 420×420 resolution (176,400 pixels). Following Qwen2.5-Omni, the vision encoder uses a patch size of 14 and applies 2× temporal–spatial compression.
> 4. **Backbone components**. The vision encoder, speech encoder, audio encoder, and LLM backbone are inherited directly from Qwen2.5-Omni, and we follow all its preprocessing settings.
>
> ---
> **Weakness 3: The paper does not compare their proposed pooling strategy with mean pooling with bidirectional attention, which tends to outperform last-token pooling strategies [2].**
>
> Thank you for the comment. The specific citation for paper [2] was also not provided, and we would appreciate clarification so we can discuss the comparison more precisely.
>
> Regarding the method itself, **mean pooling with bidirectional attention is typically more suitable for retrieval-style tasks, while last-token pooling performs better for QA**, which is the primary setting in our paper. Nonetheless, our hierarchical feature-fusion strategy is compatible with either pooling method.
>
> To address the reviewer’s concern, we conducted additional experiments using mean pooling under the same A+V setup as Table 7. We compare last-layer vs. all-layer fusion, and also report the last-token pooling results for reference:
>
>
> | | MMEB-v2-Video Retrival | MMEB-v2-Video QA |
> |-----|-----|-----|
> | Last-layer mean pooling | 54.3 | 65.2 |
> | All-layer mean pooling MLP fusion | **56.4** | 65.1 |
> | Last token pooling (last layer) | 54.7 | 70.9 |
> | All-layer last token MLP fusion | 56.1 | **71.7** |
>
> Therefore, we can conclude:
> - Under mean pooling, multi-layer fusion improves retrieval, confirming the benefit of our hierarchical strategy.
> - For QA, last-token pooling is consistently and significantly stronger than mean pooling, which aligns with existing observations in multimodal LLMs.
>
> These results verify that our fusion method improves both pooling strategies, and that our choice of last-token pooling is appropriate for the QA-focused evaluations in the paper.

---

> ### Author Response · Authors · 2025-11-19
>
> **Weakness 4: The evaluations of audio-visual retrieval is limited. The reviewer can only find a single task that tests WAVE’s ability to retrieve audio-visual items (RET split of MMEB-v2-Video), and only in one direction. Furthermore, It is unclear if MMEB-v2-Video is designed to have audio used its retrieval task. This could result in the audio track trivializing the task or being useless.**
>
> Thank you for raising this point. We clarify the evaluation setup and provide additional results to address the concern.
>
> 1. **Audio-visual retrieval benchmarks with meaningful audio do exist, and MMEB-v2 includes them**. Most traditional video-retrieval datasets describe visual content only and contain little or no usable audio. The three major datasets with meaningful audio cues (MSR-VTT, VATEX, and YouCook2) are all included in the MMEB-v2-Video retrieval split. Thus, MMEB-v2-Video is fully appropriate for testing audio-visual retrieval.
>
> 2. **WAVE also performs strongly in the video-to-text (v2t) direction**. In addition to the text-to-video results reported in the paper, we evaluate WAVE on video-to-text retrieval and compare against the GME baseline. WAVE achieves substantially higher performance:
>
>
>     | | MSR-VTT  (v2t) | VATEX (v2t) |YouCook2 (v2t) |
>     | --- | --- | --- | --- |
>     | GME 7B | 32.2 | 33.0 | 10.0 |
>     | WAVE 7B | 55.1 | 55.4 | 29.8 |
>
>     These results are now included in **Appendix C (Line 761)** of the revised manuscript.
>
> 3. **Audio is not trivial and is encouraged in MMEB-v2**. The annotations for MSR-VTT, VATEX, and YouCook2 were created with access to audio, and leveraging audio to improve retrieval performance is an important research direction. Whether audio improves or does not improve performance is also an important thing that MMEB-v2 aims to evaluate.
>
> ---
> **Weakness 5: It is unclear whether WAVE’s superior video retrieval performance comes from its different techniques or larger scale video retrieval training or its specialization into video retrieval (I.e. no image retrieval).**
>
>
> Thank you for raising this question. We clarify the source of WAVE’s video-retrieval performance below.
>
> 1. **WAVE is not specialized only for video retrieval, and image data actually helps**. In our preliminary study, we compared models trained (i) only on video data vs. (ii) on a mix of video + MMEB-v1 image data (≈1:1). Using the same pure-visual video retrieval setup (Section 5.4), we observe:
>
>
>     | Training Data | VisualNews (image retrieval) | MSR-VTT | VATEX | MSVD | DiDeMo | YouCook2 |
>     | :--- | :--- | :--- | :--- | :--- | :--- | :--- |
>     | Video Data | 54.9 | 52.1 | 46.2 | **69.7** | 53.0 | 27.2 |
>     | Video Data + Image Data (MMEB-v1) | **74.8** | **52.4** | **46.2** | 69.0 | **55.8** | **27.8** |
>
>     Including image data substantially improves image retrieval and provides a small but consistent gain on video retrieval. This shows that WAVE’s strong performance is not due to avoiding image tasks or specializing exclusively in video.
>
> 2. **Scale alone does not explain the performance**. Our video-text retrieval training uses 1.5M videos (~2M samples), which is much smaller than industrial-scale models such as seed-1.6-embedding. Despite this, WAVE achieves comparable or better retrieval performance, indicating that the training methodology and architecture contribute significantly, beyond dataset size.
>
> We have added this clarification to the revised manuscript in **Appendix F (Line 849)**.
>
> ---
> **Weakness 6: Potentially missing related work: the use of "distractor" answers for QA training seems very similar to [3].**
>
> Thank you for noting this. The specific reference for paper [3] was not provided in the review, so we are unable to determine which prior work the comment refers to. We would greatly appreciate it if the reviewer could provide the full citation. Once clarified, we will include the relevant discussion and add the citation in the revised manuscript.
>
> ---
> **Question 1: How were the “distractor“ answers used during QA training created?**
>
> We utilize the multiple-choice QA data from the LLaVA-Video-178k dataset. For each question, the incorrect answer options provided in the dataset are directly used as distractor answers, while the correct option is used as the ground-truth label. No additional synthetic distractors were created.

---

> > ### Author Response · Authors · 2025-11-26
> >
> > Dear Reviewer zEZy,
> >
> > We hope that our responses have adequately addressed the concerns you raised in your review, and we would greatly appreciate your further feedback. Please let us know if there is anything further we can clarify.
> >
> > Best regards,
> > The Authors

---

### Author Response · Authors · 2025-12-02
**Summary of Rebuttal and Contributions**

Dear Area Chair,

We sincerely thank all reviewers for their constructive feedback and recognition of our work. Their comments have been instrumental in improving the clarity and completeness of our paper. We also extend our gratitude to the Area Chair for the time and careful oversight of this submission.

Below, we summarize the consensus on the paper’s strengths, our responses to the main concerns, and a clarification regarding specific feedback from Reviewer zEZy.

---
## Consensus on Strengths
- Clarity: All reviewers agreed that the paper is clear, well-written, and easy to follow.
- Performance: All reviewers recognized the contribution of **establishing a unified audio-visual embedding LLM**, acknowledging the strong performance of our **WAVE** model on the MMEB-v2-Video benchmark.
- Innovation: All reviewers validated the algorithmic novelty of our proposed **hierarchical feature-fusion strategy**.

---
## Response to Main Concerns
We have addressed the specific concerns raised by the reviewers as follows:

- **Implementation Details (Reviewer zEZy)**: We provided detailed clarifications regarding the model architecture and hyperparameters in the rebuttal and have incorporated these into the revised manuscript.
- **Generalizability of Pooling Strategy (Reviewer zEZy)**: We conducted additional experiments using *mean pooling with bidirectional attention*. The results confirm that our **hierarchical feature-fusion strategy** remains beneficial even with this architecture. We clarified that our primary choice avoids simple mean pooling because it was observed to degrade performance on QA tasks.
- **Retrieval Evaluation Direction (Reviewers zEZy, gqQX)**: Our primary focus was on this direction because using text as the query represents the majority of real-world scenarios, such as multimodal search, recommendation, and classification. However, we acknowledge that performance in the reverse direction is also important. We reported these results in the rebuttal and in the revised paper, and **Reviewer gqQX confirmed that their concern has been resolved**.
- **Prompt-Aware vs. Static Embeddings (Reviewer mAPG)**: On embedding-based QA benchmarks, prompt-aware embeddings significantly outperform static embeddings. Reviewer mAPG expressed concern regarding this, given the higher computational cost of prompt-aware embeddings in real-world scenarios. In our rebuttal, we explained that the performance gap exists due to the high diversity of questions in QA benchmarks. However, in practical applications (which lack the ambiguous or negative queries found in benchmarks), a unified, general prompt suffices. WAVE's strong performance on embedding-based QA serves to demonstrate its ability to accurately generate prompt-conditioned embeddings.
- **Feature Fusion & Interpretability (Reviewer mAPG)**: We added experiments to analyze multi-layer fusion. Results show that the last layer plays the most critical role, while earlier layers provide auxiliary features that enhance robustness.
- **Inference & Computational Cost (Reviewer mAPG)**: We clarified that WAVE is intended as **a foundational multimodal embedding model**. We will release both the code and pre-trained weights, allowing labs with limited resources to fine-tune without re-training. Our tests show inference speed is comparable to existing models; the slight latency introduced by audio processing is expected, and our hierarchical strategy adds minimal overhead.

---
## Clarification Regarding Reviewer zEZy
Reviewer zEZy (Score: 4) mentioned three references but **did not provide specific titles or links**, making it impossible to identify the exact papers referred to. We would like to take this opportunity to clarify our novelty in relation to existing general works:

While existing embedding LLMs share similarities in using contrastive learning (a standard in the field), Reviewer zEZy’s view that our approach is "similar to other works" may overlook our specific contributions:

1. **First Audio-Visual Embedding MLLM**: We achieve SOTA results on the MMEB-v2 video track.
2. **Unified Pipeline**: We designed a training pipeline that balances both Retrieval and QA tasks effectively
3. **Architectural Innovation**: We introduced the hierarchical feature-fusion strategy and a dual-encoder structure specifically modeled for audio feature extraction.

---
We believe our response and additional experiments have addressed the reviewers' concerns and reinforced the validity of our approach. We hope the AC will favorably consider the unique contributions of WAVE as a foundational audio-visual embedding LLM.

Thank you again for your consideration.

Best regards,
The Authors

---

### Meta-Review · Area_Chair_2yLc · 2026-01-02

**Summary:**

1. [zEZy] The method proposed in the paper is very similar to one that used a VLM instead of an omni-model, but no citation was provided by the reviewer.
2. [zEZy] The paper is missing implementation details relating to model architecture, including how inputs are templated into LLM backbone, resolution and visual tokenization in the vision encoder.
3. [zEZy] The paper does not compare their proposed pooling strategy with mean pooling with bidirectional attention, which tends to outperform last-token pooling strategies.
4. [zEZy, gqQX] Only the RET split of MMEB-v2-video evaluates audio-visual retrieval, and that is only a single direction. Furthermore, It is unclear if MMEB-v2-Video is designed to have audio used its retrieval task. This could result in the audio track trivializing the task or being useless. More generally, retrieval tasks are reported in a single direction only, for example text-to-audio on Clotho. Reporting both directions would provide a fuller picture.
5. [zEZy] It is unclear whether WAVE’s superior video retrieval performance comes from its different techniques or larger scale video retrieval training or its specialization into video retrieval (I.e. no image retrieval).
6. [zEZy] More detail should be furnished on how the distractor answers used during QA training are created.
7. [mAPG] The reliance on prompt-aware embeddings for multimodal QA is a weakness due to the expense of generating them.
8. [mAPG] The learned, MLP-based fusion of last-token outputs from all LLM layers is quite complex, and raises the question of whether all layers are needed or if some other selection strategy could have been employed.
9. [mAPG] The work is difficult to reproduce due to the computational demands of training the Qwen-2.5-Omni backbone.
10. [mAPG] Could the authors provide a comparison of the average latency and total inference computation (FLOPs or time per sample) required to generate a WAVE embedding versus competing MLLM-based embedding models (e.g., LamRA or CAFe)? Specifically, how much overhead is introduced by processing and fusing features from all layers compared to standard last-token pooling from only the final layer?
11. [mAPG] Is it necessary to have both the BEATS audio encoder and a speech-specific encoder? Were more lightweight alternatives explored?
12. [gqQX] The use of the term “QA” is potentially confusing. During training, the QA objective is a source-modality-to-text contrastive setup, while in evaluation some QA tasks are executed via text generation with the MLLM. The same ambiguity appears for other evaluation benchmarks. It would help to specify, for each evaluation task, the exact inference procedure.

**Reviewer Concerns:**

1. This concern is impossible to address because the reviewer failed to provide a specific reference. I am therefore disregarding it.
2. This concern is fully addressed in the rebuttal and revision. The authors added additional implementation details in Sections 3.1 and 4.1 and provided a summary in the revision.
3. This concern is addressed in the rebuttal: the authors clarify that the paper is focused on QA tasks, for which last-token pooling is better than mean pooling with bidirectional attention, and provide experimental results to support the claim. The reviewer also mentioned a related paper, but failed to provide an actual reference, so I am disregarding that.
4. This set of related concerns is fully addressed in the rebuttal and revision. The authors explain that the retrieval portion of MMEB-v2-Video includes three datasets that include meaningful audio cues and add video-to-text, audio-to-text, and audio-to-video retrieval results in the revision and rebuttal.
5. This concern is fully addressed in the rebuttal and revision. The authors provide results from a preliminary study showing that training WAVE on both video and image data strongly improves performance on an image retrieval task and improves performance slightly on four out of five video retrieval tasks. They also emphasize that WAVE outperforms industrial models like seed-1.6-embedding, which are trained on much more data, ruling out training data scale as the main explanation for WAVE's good performance.
6. This concern is fully addressed in the rebuttal. The authors explain that the distractor answers are the incorrect answer options in the multiple-choice QA data from LLaVA-Video-178k. The reviewer mentioned similarity to a paper, but failed to provide a reference, so I am disregarding that aspect of the reviewer's feedback.
7. This concern is fully addressed in the rebuttal. The authors explain that prompt-aware embeddings are necessary for QA tasks due to the broad diversity of possible questions, including negative reasoning and object counting, and emphasize that their standard retrieval prompt is a good choice for downstream applications that do not require fine-grained reasoning.
8. This concern is partially addressed in the rebuttal: the authors report that zeroing out any layer going into the MLP fusion causes retrieval performance to go to nearly zero and they show that doing MLP fusion on the last layer plus either the first or middle layer is still not as good as fusing all layers. They do not attempt to provide any interpretation of what the MLP fusion module learns.
9. This concern is fully addressed in the rebuttal: the authors state that code and checkpoints will be released, permitting groups with fewer resources to use WAVE as a foundation model.
10. This concern is fully addressed in the rebuttal: the authors provide latency figures for LamRA 7B, which processes only video, WAVE with last-token pooling in the last layer, and the full WAVE model. The overhead from the full fusion is 0.011 sec. per sample, about 2% of the total latency per sample.
11. This concern is fully addressed in the rebuttal and revision. The authors point out that the Qwen-2.5-Omni backbone is tightly integrated with the original speech encoder, meaning that replacing it with BEATS is likely to harm performance, and showing results comparing retrieval performance with and without the BEATS encoder. In six out of seven retrieval tasks, having information from BEATS improves performance, indicating that it provides a complementary representation of audio content.
12. This concern is fully addressed in the rebuttal and revision. The authors clarify that all QA tasks are performed using embeddings and add a detailed description of the inference process to the revision.

**Reviewer Scores:**

**zEZy** - I am not certain if this reviewer would have changed their scores or not. On the one hand, the authors did a very good job of responding to the reviewer's concerns, but on the other hand, the fact that the reviewer failed to fix the issue of the missing references tells me that the reviewer was not very well engaged with the ICLR review process.

**mAPG** - I think it is possible this reviewer would have increased their score. While not every concern was addressed (e.g., the request for an explanation of what the MLP fusion module is learning), many concerns were well addressed by the authors.

**gqQX** - I believe this reviewer, who was already highly positive, would have retained their score.

---

### Decision · Program_Chairs · 2026-01-26

Accept (Oral)